# Phenotypic proteomic profiling identifies a landscape of targets for circadian clock–modulating compounds

Sandipan Ray[1,2] , Radoslaw Lach[3], Kate J Heesom[4], Utham K Valekunja[1,2], Vesela Encheva[3] , Ambrosius P Snijders[3], Akhilesh B Reddy[1,2]

**Determining the exact targets and mechanisms of action of drug molecules that modulate circadian rhythms is critical to develop novel compounds to treat clock-related disorders. Here, we have used phenotypic proteomic profiling (PPP) to systematically determine molecular targets of four circadian period–lengthening compounds in human cells. We demonstrate that the compounds cause similar changes in phosphorylation and activity of several proteins and kinases involved in vital pathways, including MAPK, NGF, B-cell receptor, AMP-activated protein kinases (AMPKs), and mTOR signaling. Kinome profiling further indicated inhibition of CKId, ERK1/2, CDK2/7, TNIK, and MST4 kinases as a common mechanism of action for these clock-modulating compounds. Pharmacological or genetic inhibition of several convergent kinases lengthened circadian period, establishing them as novel circadian targets. Finally, thermal stability profiling revealed binding of the compounds to clock regulatory kinases, signaling molecules, and ubiquitination proteins. Thus, phenotypic proteomic profiling defines novel clock effectors that could directly inform precise therapeutic targeting of the circadian system in humans.**

## Introduction

Circadian clocks have a profound impact on human health as they play a central role in coordinating daily physiological and behavioral processes. Circadian clocks, sleep–wake cycles, and metabolic networks systematically interact with each other to maintain a cycling pattern in cellular functions, activity and energy utilization in accordance to a daily (near 24-h) rhythm (Bass & Takahashi, 2010; Asher & Schibler, 2011; Ray & Reddy, 2016). In mammalian circadian organization, the brain's suprachiasmatic nucleus (SCN) acts as the master clock and orchestrates synchronization of oscillators in peripheral tissues (Yamazaki et al, 2000; Reppert & Weaver, 2002). Circadian misalignment or desynchrony due to old age, neurological

disease, shift work, travel across time zones, and irregular food intake is a rising cause of morbidity. Circadian dysfunction is associated with an increased risk of cancers (Hoffman et al, 2008; Papagiannakopoulos et al, 2016), diabetes (Marcheva et al, 2010; Pan et al, 2011), cardiovascular, and metabolic disorders (Scheer et al, 2009; Buxton et al, 2012). Consequently, small molecule compounds capable of modulating circadian rhythms are promising for therapeutically targeting diverse types of human disease linked to circadian dysregulation (Chen et al, 2018).

High-throughput screenings of small molecules have identified a few chemical compounds that can modulate circadian rhythm period length in a dose-dependent manner (Hirota et al, 2008, 2010; Chen et al, 2012; Tamai et al, 2018; Lee et al, 2019; Oshima et al, 2019). Protein kinases, including casein kinase 1 (CK1 d/e), casein kinase 2 (CK2), JNK, glycogen synthase kinase 3-$\beta$ (GSK3-$\beta$), and CDKs, are considered as the possible targets for altering circadian clock period (Hirota et al, 2008, 2010; Isojima et al, 2009; Walton et al, 2009; Yagita et al, 2009; Oshima et al, 2019). However, these mechanisms have not been illustrated in a comprehensive and comparative manner, as well as the cellular effects of these compounds. In particular, whether period-altering compounds share as yet undescribed common downstream pathways remains an open question. Defining such targets would enable precise targeting in circadian drug discovery campaigns, which is currently lacking because of a gap in our knowledge concerning the mechanisms of action of circadian-active compounds.

In recent years, mass spectrometry (MS)–based quantitative proteomics, more particularly thermal proteome profiling (TPP), has emerged as a powerful approach in decoding molecular mechanisms and cellular targets for novel or existing drugs in a more inclusive and unbiased fashion (Franken et al, 2015). The major advantage of the TPP methodology is the capability to measure target occupancy of drugs by screening thousands of proteins in parallel in living cells or tissues (Martinez Molina et al, 2013; Savitski et al, 2014). Here, we applied an approach that we term phenotypic proteomic profiling (PPP), integrating multipronged proteomics approaches, including global proteome, phosphoproteome, kinome mapping, and proteome-wide profiling of thermal stability (TPP) to decipher the molecular targets

[1]Department of Systems Pharmacology and Translational Therapeutics, Perelman School of Medicine, University of Pennsylvania, Philadelphia, PA, USA  [2]Institute for Translational Medicine and Therapeutics, Perelman School of Medicine, University of Pennsylvania, Philadelphia, PA, USA  [3]The Francis Crick Institute, London, UK  [4]Proteomics Facility, University of Bristol, Bristol, UK

Correspondence: areddy@cantab.net; sandipan.ray@cantab.net

of four circadian period–lengthening compounds (longdaysin, pur-valanol A, roscovitine, and SP600125) in human osteosarcoma U2OS cells—a robust and well-characterized circadian model system (Fig 1). We report a comprehensive landscape of effector proteins for the compounds that impact diverse aspects of cellular physiology. Importantly, we determined several common molecular targets and physiological effects of the circadian rhythm–modulating compounds. Identification of such targets will form the basis for the future development of new drugs that can target the clockwork in a precise manner.

## Results

### Circadian period–lengthening compounds reshape the proteome and phosphoproteome

Small molecule screens using circadian bioluminescence reporter cell lines have identified a handful of circadian period–lengthening compounds (Hirota et al, 2008, 2010). We focused here on four compounds—longdaysin, purvalanol A, roscovitine, and SP600125—for in-depth mechanistic analysis because they all result in the same circadian clock phenotype (period lengthening). As such, we hypothesized that these compounds might alter cellular physiology in a similar way and share common downstream pathways. We first validated that the compounds alter circadian period length by assaying bioluminescence rhythms of *Per2-dLuc* U2OS cells. Longdaysin

drastically lengthened circadian period (35.1 ± 1.29 h, 8 $\mu$M), whereas roscovitine (28.9 ± 0.08 h, 8 $\mu$M), SP600125 (28.6 ± 0.1 h, 7 $\mu$M), and purvalanol A (25.2 ± 0.04 h, 8 $\mu$M) caused significant, but less pronounced, dose-dependent increases in the period length (Fig S1).

We next defined the effects of the circadian rhythm–modulating compounds on the cellular proteome by using a multiplexed tandem mass tag (TMT)–based quantitative approach (Fig 2A). Such multiplexing using stable isotope labelling results in increased throughput, higher precision, better reproducibility, reduced technical variation, and fewer missing values than other methods (Mertins et al, 2018; O'Connell et al, 2018). Indeed, we found that reproducibility between the biological replicates of control (DMSO only) and compound-treated samples was excellent (Pearson R > 0.95) (Fig S2A). There was a slight difference in circadian phases in the treated samples (particularly for longdaysin) compared with the vehicle-treated control cells because of the period-lengthening effects of these compounds (Fig S1). Consequently, to specify the proteins that are altered directly by these compounds, not because of the difference in circadian phases, we excluded the rhythmic candidates in U2OS cells as reported earlier by us (Rey et al, 2016) and others (Hughes et al, 2009). However, we identified only three (Pacsin3, Tgm2, and Shc1) such overlapping candidates. Comparative proteomic analysis of treated (10 $\mu$M) versus control cells indicated the increased abundance of kinase inhibitors and histone proteins and reduced levels of CDKs and other kinases after treatment with the compounds (Table S1). On a global scale, we observed alterations in several classes of proteins with some overlapping candidates (Fig S2B and C). Interestingly, histone proteins

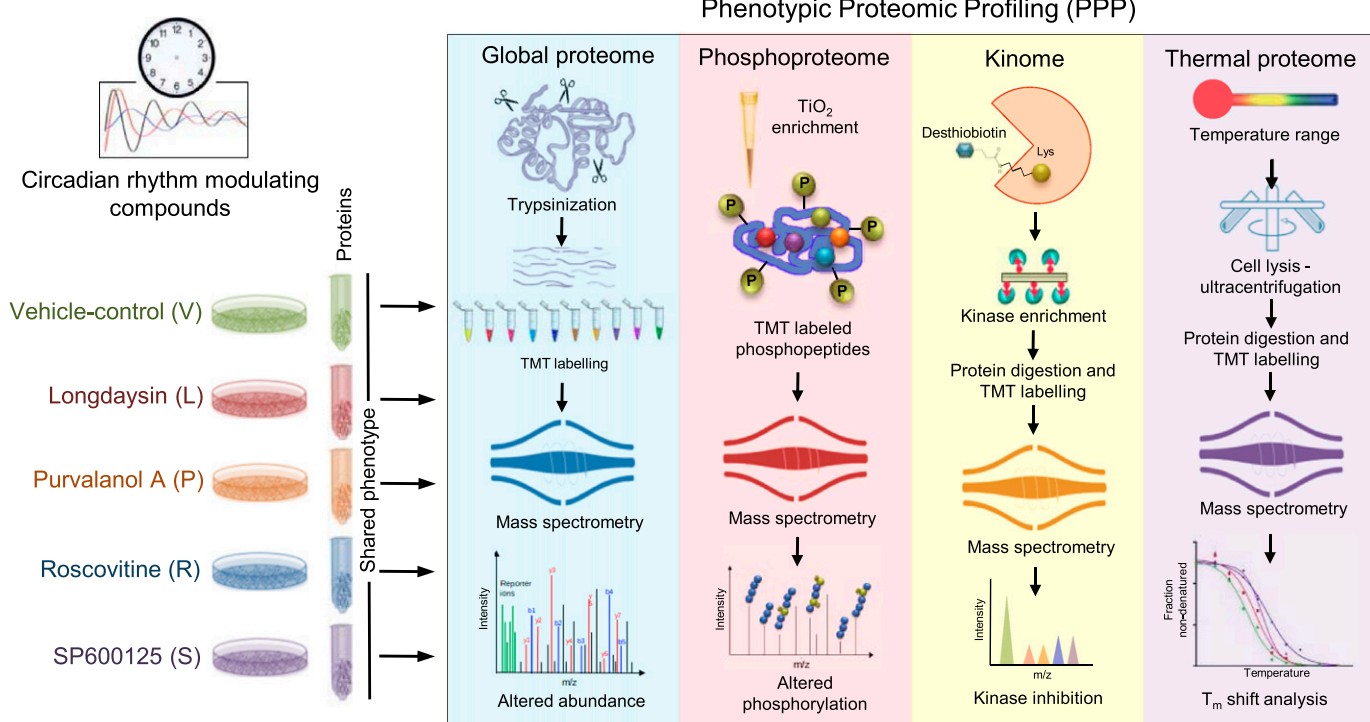

**Figure 1. PPP for the identification of common targets for circadian period–altering compounds.**
Schematic showing PPP analysis of circadian period–altering compounds. A multidimensional quantitative proteomics analysis pipeline integrating global proteome, phosphoproteome, kinome mapping, and proteome-wide profiling of thermal stability.

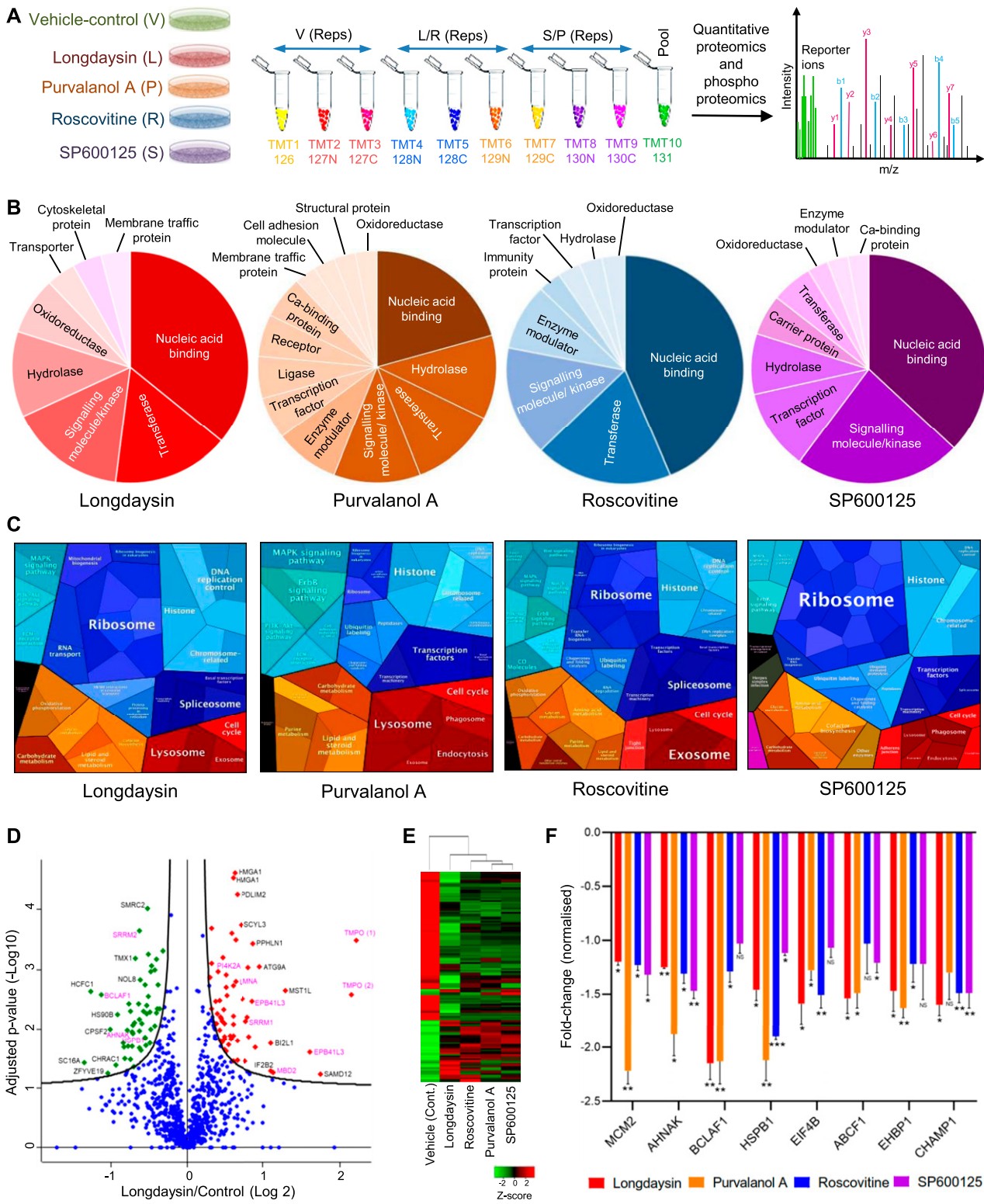

**Figure 2. Circadian period–lengthening compounds induced alterations in U2OS proteome and phosphoproteome.**
**(A)** Schematic representation showing the labelling strategy used in TMT 10-plex quantitative proteomics and phosphoproteomics experiments. **(B)** Pie charts showing functional classification (protein class) for the differentially abundant proteins ($p_{Adj}$ < 0.05) identified in longdaysin-, purvalanol A–, roscovitine-, and SP600125-treated cells. **(C)** Proteomaps (KEGG Pathways gene classification) of the differentially abundant proteins identified in cells treated with circadian period–lengthening compounds. **(D)** Volcano plots showing *P*-values (–log10) versus ratio of group mean of phosphosites abundance (log2) in control (DMSO) and longdaysin-treated cells. Red, up-regulated; green, down-regulated (adjusted *P*-value < 0.05), and blue, not significantly altered phosphosites. Selected differentially abundant phosphosites are labelled. Phosphosites labelled in pink

(H3 and H4) were consistently altered across all the compound-treated conditions (Fig S2C). Importantly, rhythmic acetylation/deacetylation of histones (H3 and H4) at multiple clock target genes plays crucial roles in the mammalian circadian clock (Etchegaray et al, 2003). Differentially abundant proteins upon treatment with circadian rhythm–modulating compounds were largely nucleic acid–binding proteins, kinases, signaling molecules, ribosomal proteins, enzyme modulators, and transcription factors (Fig 2B and C). These proteins are associated with diverse, vital, physiological pathways, including translation, signal transduction cascades, DNA damage and methylation, and metabolism (Fig S2D).

Reversible protein phosphorylation plays a central role in circadian rhythmicity (Virshup et al, 2007) and rhythmic tuning of signaling pathways (Robles et al, 2017). Therefore, we asked whether the circadian rhythm–modulating compounds change the phosphorylation pattern of the cellular proteome by using multiplexed quantitative phosphoproteomics. Comparison of the cellular phosphoproteomes demonstrated altered mean abundance (against control-vehicle only) of multiple phosphosites in compound-treated cells (Fig S3A). Many of these altered phosphosites were modulated in common by multiple compounds (Figs 2D and S3B). Of note, 123 phosphosites were significantly altered ($p_{Adj} < 0.05$) by at least one of the compounds. This allowed effective separation of control versus compound-treated cells by hierarchical clustering (Figs 2E and S3C). Treatment with the compounds reduced phosphorylation of many proteins involved in intracellular signal transduction, apoptosis, the cell cycle and protein translation (Figs 2F and S3D). Importantly, some altered phosphoproteins such as MCM2, BCLAF1, and HSPB1 are substrates of clock-regulating kinases, including CKs, CDKs, and MAPKs (Montagnoli et al, 2006; Franchin et al, 2018). Collectively, the proteomics and phosphoproteomics assessments indicate modulation of a diverse range of cellular targets by these period-lengthening compounds.

## Circadian period–lengthening compounds inhibit several kinases

It is widely recognized that protein kinases serve as key regulators of mammalian circadian clocks (Reischl & Kramer, 2011). CK1 d/e, CK2, AMP-activated protein kinases (AMPKs) and many other protein kinases regulate phosphorylation, functional activity, complex formation and degradation of the core transcriptional clock proteins including PER, CRY, CLOCK, BMAL1 and REV-ERB (Eide et al, 2002; Lamia et al, 2009; Lipton et al, 2015; Tamaru et al, 2009; Narasimamurthy et al, 2018). CK1 also has regulatory effects in temperature compensation of non-transcriptional oscillators in human red blood cells (Beale et al, 2019). Consequently, small-molecule kinase inhibitors are considered as the most promising candidates for tuning circadian period in mammalian clocks for therapeutic benefit (Yagita et al, 2009). To this end, the four circadian period–altering compounds we studied here are also known kinase inhibitors–longdaysin (CK1alpha/delta and ERK2 inhibitor) (Hirota et al, 2010), purvalanol A (CDK inhibitor) (Bain et al, 2007), roscovitine (CDK and ERK inhibitor) (Bain et al, 2007; Whittaker et al, 2018), and SP600125 (JNK and CDK inhibitor) (Bennett

et al, 2001). However, comprehensive profiling of the whole kinome with these compounds has not been investigated earlier, which is crucial for the identification of their common and novel targets.

Desthiobiotin nucleotides can selectively enrich and profile target enzyme classes to evaluate the specificity and affinity of diverse enzyme inhibitors (Patricelli et al, 2007; Cravatt et al, 2008) (Fig S4A). MS can then be used for kinome analysis using such nucleotide probes, allowing global delineation of the regulated kinases in treated cells (McAllister et al, 2013). We used desthiobiotin-ATP enrichment in combination with multiplexed quantitative proteomics to measure kinase abundances in human U2OS cells treated with circadian period–lengthening compounds (Fig 3A). We obtained more than 85% enrichment of the desthiobiotin-ATP peptides (Fig S4B), which allowed quantification of many extremely low abundance kinases that are generally not detected in regular quantitative proteomics workflows. We measured abundance and inhibition profiles for 174 kinases and observed reduced activity (major, moderate, or minor effects) of several kinases induced by the compounds (Fig S4C). Of note, a number of kinases, including CK1d, ERK1/2, CDK7, CDK2, CAMK1d/2d, MST4/STK26, and TNIK were detected as overlapping targets for the compounds (Fig 3B). To this end, CK1a/d/e and ERK2 were identified earlier as the targets responsible for period-lengthening in mammalian cells (Isojima et al, 2009; Hirota et al, 2010).

We next validated the MS findings by using radiometric "HotSpot" assays. This assay directly measures kinase catalytic activity towards a specific substrate (Fig S5A) and is a well-established method for kinase inhibition analysis (Anastassiadis et al, 2011; Duong-Ly et al, 2016). Radiometric assays similarly demonstrated inhibition of multiple kinases by the circadian rhythm–modulating compounds, with a few overlapping targets (Fig S5B). Importantly, we saw functional inhibition (>50%) of CK1d, CDK7, CDK2, and MST4 by all of these compounds (Fig 3C), with SP600125 inhibiting the maximum number of kinases (Fig S5B). We further measured half-maximum inhibitory concentration ($IC_{50}$) for the compounds towards five overlapping target kinases (CK1d, CDK2, CAMK2d, MST4, and TNIK) using a 10-dose inhibition series (Fig 3D). The most affected kinases were CK1d and CDK2 with an $IC_{50} < 5\ \mu M$ for all four compounds, whereas purvalanol A and SP600125 inhibited TNIK specifically with a more potent $IC_{50}$ (<0.5 $\mu M$) (Fig S6).

Taken together, the results from desthiobiotin-ATP enrichment and radiometric assays suggest that the circadian rhythm–modulating compounds exhibit their effects by inhibiting multiple protein kinases. Therefore, we examined the distribution of the identified kinase targets in their families. We found that the compounds inhibited kinases from all major typical kinase subfamilies, with the highest number of targets (10) from CMGC (CDKs and MAPKs) and calmodulin/calcium-regulated kinase (CAMK) subfamilies (Fig 3E). Multiple members of automatic gain control (AGC) and STE subfamilies were also targets of the compounds. Importantly, several members of the CMGC (ERK2/MAPK1, CDK7, and CDK2), CAMK (CAMK1d and 2d), STE (MST4 and TNIK), and CK (CK1d) subfamilies were overlapping targets for the compounds (Fig 3F).

---

were also altered by other circadian period–altering compounds. **(E)** Hierarchical cluster analysis of 123 phosphosites that are altered by circadian period–lengthening compounds. **(F)** Relative abundance (against DMSO control) of the commonly altered phosphoproteins in longdaysin-, purvalanol A–, roscovitine-, and SP600125-treated cells. Data are represented as mean ± SEM (n = 3). One-way ANOVA, Dunnett's test. *** indicates $P < 0.0001$, ** indicates $0.0001 < P < 0.001$, * indicates $0.001 < P < 0.01$, and NS indicates $P > 0.05$.

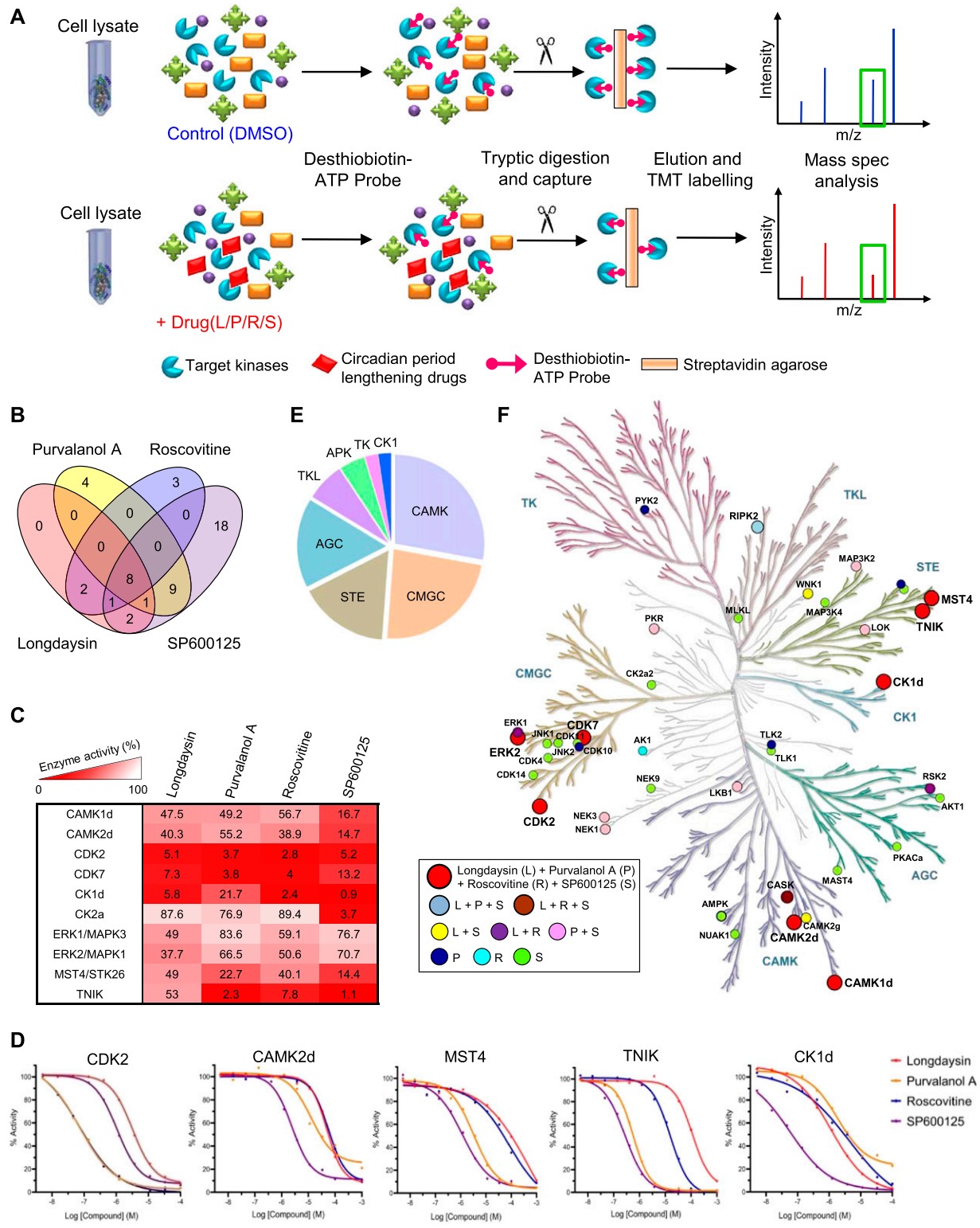

**Figure 3. Circadian period–altering compounds inhibit several kinases in the mammalian system.**
**(A)** Schematic representation showing kinase profiling of U2OS cell line in the presence of circadian period–lengthening compounds (longdaysin, purvalanol A, roscovitine, and SP600125) by using desthiobiotin nucleotide probes in combination with TMT-based quantitative proteomics (biological replicates, n = 3). This workflow allows targeted capture and quantification of kinases using the active-site probes (desthiobiotin-ATP). **(B)** Venn diagram representing the overlap between the kinases inhibited ($p_{Adj}$ < 0.05) by different circadian period–lengthening compounds. **(C)** Activity profiles of different kinases in the presence of circadian period–lengthening compounds (100 $\mu$M, n = 3) measured by a radiometric "HotSpot" assay. **(D)** 10-dose inhibition curves (threefold serial dilution starting at 100 $\mu$M) representing overlapping target kinases for different circadian period–lengthening compounds. **(E)** Distribution of the identified kinase targets of the circadian period–altering

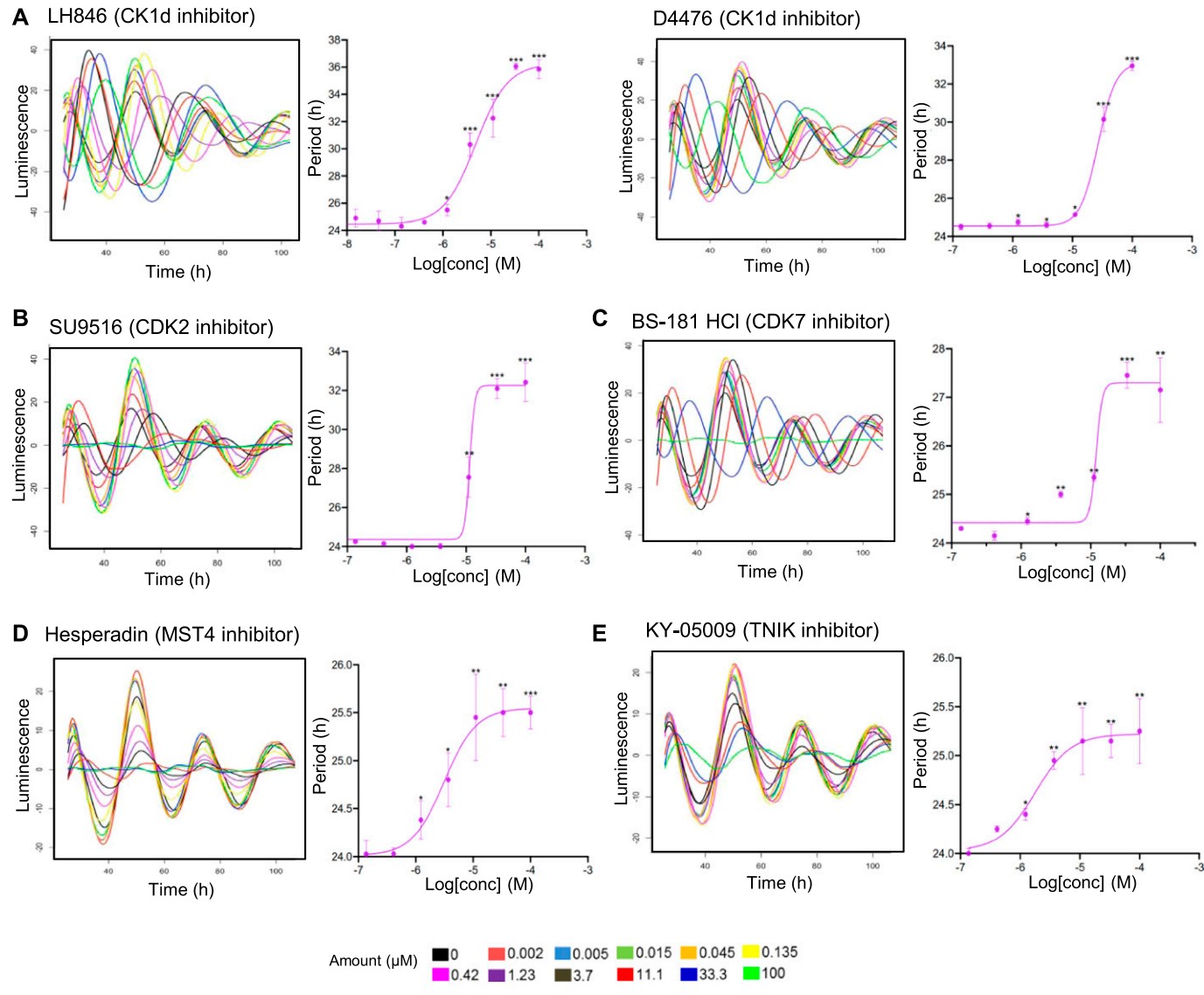

**Figure 4.  Dose-dependent effects of kinase inhibitors in mammalian circadian period length.**
**(A, B, C, D, E)** Dose-dependent effect for inhibitors of (A) casein kinase I (CK1d)—LH846 and D4476, (B) CDK2—SU9516, (C) CDK7—BS-181 HCl, (D) serine/threonine-protein kinase 26 (MST4)—Hesperadin, and (E) TRAF2 and NCK-interacting protein kinase (TNIK)—KY-05009 were evaluated. Luminescence rhythms of *Per2-dLuc* U2OS cells were monitored in the presence of 11 different concentrations of these six kinase inhibitors (threefold serial dilution starting at 100 $\mu$M). Period parameter was obtained by the curve fitting and plotted against the final concentration of the compounds (adjusted by keeping the control *Per2-dLuc* cells period length as 24 h). Dose–response curves showing the circadian period length alterations with the top seven concentrations of the compounds (100–0.14 $\mu$M). Data are the representative of multiple biological replicates (n = 4, mean ± SEM). One-way ANOVA, Dunnett's test. *** indicates $P < 0.0001$, ** indicates $0.0001 < P < 0.001$, and * indicates $0.001 < P < 0.05$.

Thus, kinome analysis defines overlapping targets for the circadian rhythm–modulating compounds, many of which were not previously known to be associated with clock function.

To establish a mechanistic connection between the identified kinase targets for the period-lengthening compounds and the clockwork, we next tested whether inhibitors of overlapping kinase targets of the compounds prolong cellular circadian rhythms. This would be expected if they were truly affecting the molecular clock in a specific manner. We used LH846 and D4476 (CK1d inhibitors), SU9516 (CDK2 inhibitor), BS-181 HCl (CDK7 inhibitor), Hesperadin (MST4 inhibitor), and KY-05009 (TNIK inhibitor) to inhibit each respective kinase. We profiled circadian rhythms of cells treated with each inhibitor by monitoring bioluminescence rhythms of *Per2-dLuc* U2OS reporter cells. Most of the inhibitors, particularly LH846, D4476, and SU9516, led to a substantial dose-dependent increase in the period length (Fig 4). Whereas LH846, D4476, and BS-181 have previously been

compounds in the kinase family. Kinase profiling was performed combining the targets obtained in quantitative proteomics and in radiometric assay. **(F)** Kinome tree illustrating the kinase targets of circadian period–altering compounds. Kinome tree structured courtesy of Cell Signalling Technology, Inc. (www.cellsignal.com) and annotated using KinMap online tool.

reported to induce period lengthening (Hirota et al, 2010; Lee et al, 2011; Ou et al, 2019), the other inhibitors were not known to have circadian period–modulating ability. Notably, we also observed dose-dependent phase-shifting effects of KY-05009, SU9516, BS-181 HCl, and D4476 (Fig S7). Such phase-shifting agents might be beneficial to treat jetlag, or aid recovery from shift work–associated health problems (Wallach & Kramer, 2015). Thus, inhibiting the kinases that we identified as the overlapping targets for the period-lengthening compounds phenocopied the effects of the compounds themselves, demonstrating their functional importance as clock modulators.

### Circadian rhythm–modulating compounds interact with clock machinery through diverse signaling pathways

We next evaluated the effects of RNAi-mediated inhibition of the kinases that were identified as overlapping targets for multiple circadian period–modulating compounds in our integrated proteomics-phosphoproteomics and kinome profiling analyses. We used data from the BioGPS circadian genomic siRNA database (Zhang et al, 2009) and determined that siRNA knockdown of these kinase targets prolongs period length in circadian oscillations (Fig S8A). Of note, knockdown of most of the kinase targets (seven of nine) with two independent siRNA pairs lengthens the circadian period: CDK2 (26.9/39.3 h), CDK7 (26/27.5 h), CK1d (26.6/27.2 h), CK2a (31.5/31 h), ERK1/MAPK3 (27.1/48 h), ERK2/MAPK1 (26.3/27.5 h), and MST4 (26.03/27.6 h) (Fig S8B).

Alterations in the activity of several kinases by the circadian modulators suggested an interaction between the molecular circadian oscillator and different signaling pathways to maintain a robust period length. We, therefore, determined the nature of these connections using pathway and network analyses. Ingenuity pathway analysis (IPA) indicated cell signaling as the top-scoring overlapping network for longdaysin (score: 41, focus molecule: 19/35), roscovitine (score: 39, focus molecule: 18/35), and SP600125 (score: 59, focus molecule: 28/35) (Fig S9). We next analysed the possible interactions among the inhibited kinases (and altered phosphoproteins) and clock components using the STRING database. We found a dense interaction network (protein–protein interaction [PPI] enrichment $P$-value = 3.04 × 10$^{-7}$) among the clock components and the targets for circadian modulator compounds (Fig 5A). Significantly, we identified the common kinase targets (CK1d, MAPKs, and CDKs) for the circadian rhythm–modulating compounds as the major connecting hubs in the network. Moreover, siRNA knockdown of many target kinases and phosphoproteins lengthened the molecular clock period (Fig 5A).

A plethora of signaling pathways were overrepresented for the long-period targets, including B-cell receptor signaling (false discovery rate [FDR] = 9.09 × 10$^{-20}$), MAPK signaling (FDR = 1.13 × 10$^{-18}$), hepatocyte growth factor signaling (FDR = 1.46 × 10$^{-16}$), NGF signaling (FDR = 2.11 × 10$^{-16}$), AMPK signaling (FDR = 3.90 × 10$^{-14}$), mTOR signaling (FDR = 7.88 × 10$^{-13}$), EGF signaling (FDR = 5.22 × 10$^{-10}$), and SAPK/JNK signaling (FDR = 2.89 × 10$^{-08}$) (Fig 5B). Furthermore, we determined that components of these signaling pathways crosstalk with each other and play vital roles in other pathways (Fig 5C). In line with this, earlier studies have demonstrated that core clock components link circadian timing and mTOR (Cornu et al, 2014;

Lipton et al, 2015), insulin (Zhang et al, 2009), MAPK (Coogan & Piggins, 2004), and AMPK (Lamia et al, 2009) signaling pathways. Taken together, our findings indicate that connections between signaling pathways form a multidimensional network that acts in concert with clock components to maintain circadian oscillations in daily physiological and metabolic processes.

### TPP defines additional cellular targets for circadian modulators

Recent studies have demonstrated that changes in the thermal stability of proteins upon ligand binding can be used to identify targets of small molecules or drugs (Martinez Molina et al, 2013; Savitski et al, 2014; Miettinen et al, 2018). TPP determines melting curves for thousands of proteins and enables potential drug targets to be found by evaluating the differential stability (change in melting temperature, $\Delta T_m$) of proteins under ligand-bound and control (unbound) conditions (Franken et al, 2015). We, therefore, applied TPP to decipher potential molecular targets of longdaysin, roscovitine, and SP600125 in U2OS cells (Fig 6A). We were able to identify 5,400 proteins in control (vehicle only) and compound-treated cells and performed melting curve analysis on nearly 75% of the detected proteins that were quantified in all the control and compound-treated samples (Fig S10A).

TPP revealed direct binding (a positive $\Delta T_m$ shift) of circadian-modulating compounds to clock regulatory kinases (CKs and CDKs) and a number of kinase regulators and signaling mediators including CD40, CD59, F-box proteins, and RNF181 (Figs 6B–D and S10–S12). Moreover, in concordance with our findings from differential quantitative proteomics, where we identified altered expression of many ribosomal proteins involved in protein translation (Figs 2C and S1D), we found $T_m$ shifts for the ribosomal proteins RPL36 and RPL37 (Figs S10 and S12). Interestingly, a recent study showed that the core circadian clock rhythmically interacts with the translational machinery via BMAL1 (Lipton et al, 2015). We did not observe (or were unable to measure) a significant $T_m$ shift for some of the kinase targets (MST4, TINK, and ERKs) that we found using desthiobiotin-ATP enrichment and radiometric assays (see above). We consider that there are three possible reasons for this (1) the proteins were affected (e.g., downstream targets); but were not direct binding partners of the circadian rhythm–modulating compounds; or (2) they were not stabilized by compound binding; or (3) they were not quantified in each MS run (without an enrichment method) because of their low abundance.

We next compared the overlapping binding targets for the compounds and identified 10 candidates that were commonly regulated (Figs 6E and F, and S13A). The magnitude of the $T_m$ shifts induced by longdaysin, roscovitine, and SP600125 was fairly correlated ($R^2 > 0.75$), indicating that the compounds bind overlapping targets with comparable affinities (Fig S13B). Many of the overlapping targets, including FBX21, RPL37, RABEP1, and CSTF2T, are associated with the protein polyubiquitination machinery and protein translation (Figs 6F and S13C). Like phosphorylation, ubiquitination, which is largely mediated by F-box proteins, also modifies clock proteins to regulate circadian timekeeping in mammals (Hirano et al, 2013; Xing et al, 2013). In particular, F-box

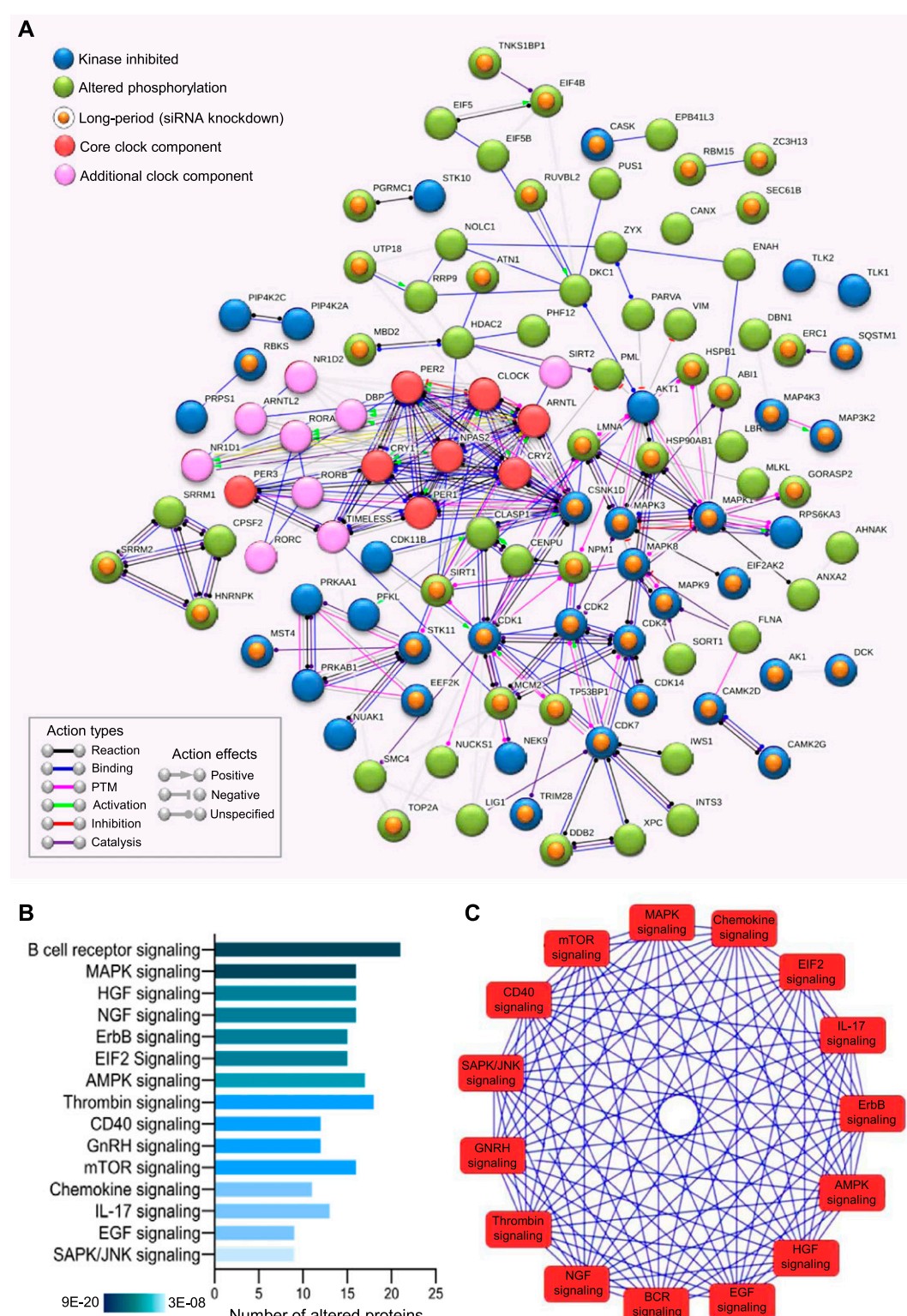

**Figure 5.  Circadian period–altering compounds target multiple signaling pathways.**
**(A)** Interaction network (PPI enrichment *P*-value 3.04 × 10$^{-7}$) among phosphoproteins (green) and kinases (blue) modulated by circadian period–altering compounds. siRNA knockdown of the candidates highlighted with a golden yellow dot induces long period length in circadian oscillations (>26 h at least in one of the two independent siRNA pairs). siRNA knockdown data obtained from BioGPS circadian layout database (http://biogps.gnf.org/circadian/) (Zhang et al, 2009). Core clock (red) and additional clock (pink) components are also integrated into the network. Association network is generated using the STRING database involving only high confidence (score > 0.8) interactions. **(B)** Signaling pathways associated with the phosphoproteins and kinases altered by circadian period–modulating compounds (analysed using IPA, top 15, FDR < 0.05). **(C)** Interaction network among the signaling pathways associated with the phosphoproteins and kinases altered by circadian period–modulating compounds.

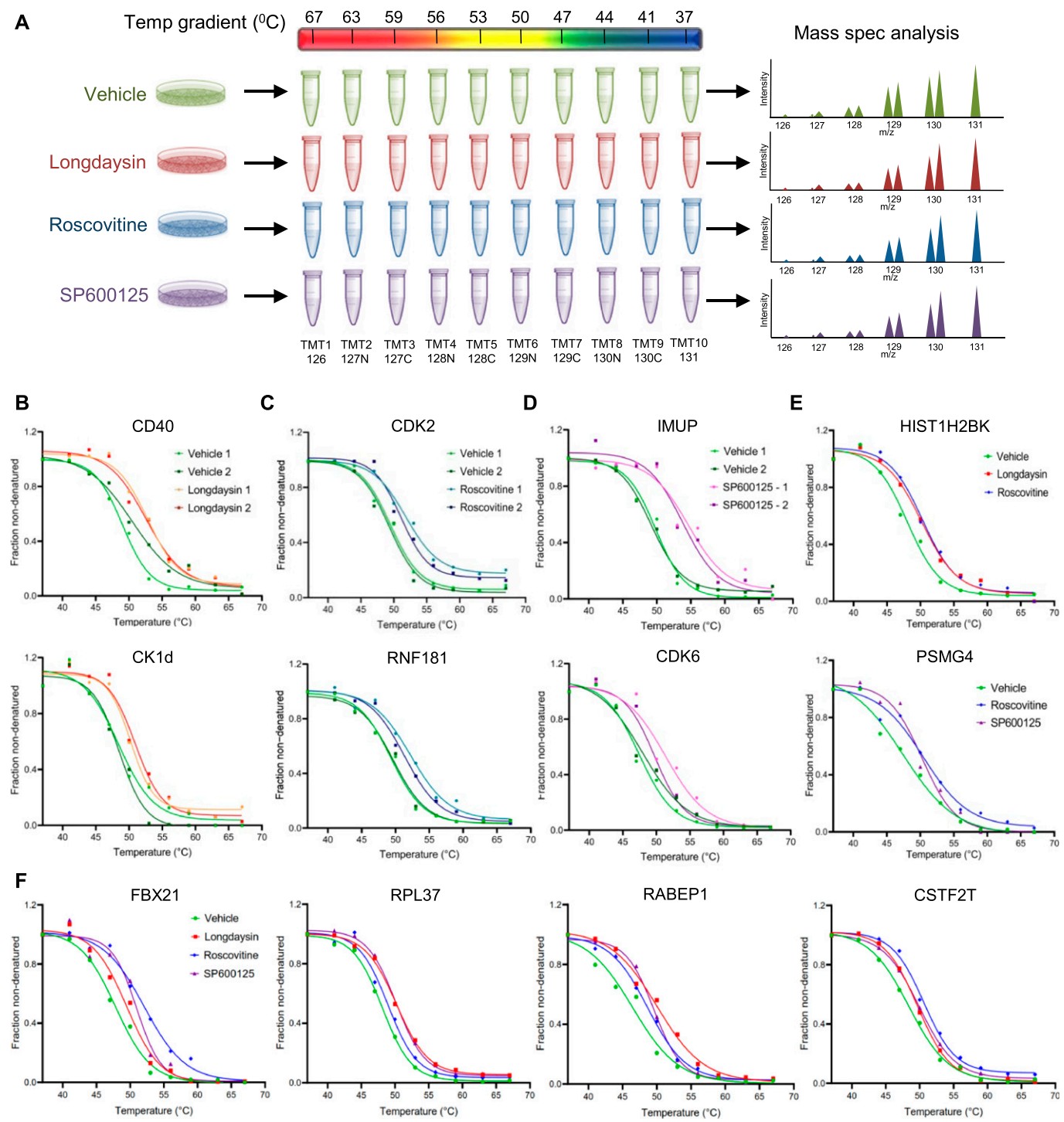

**Figure 6. Proteome-wide profiling of thermal stability after treatment with circadian rhythm–modulating compounds.**
**(A)** Schematic representation of differential profiling of the effects of three circadian period–altering compounds on the TPP in U2OS cells using TMT-based quantitative proteomics. The obtained TMT reporter ion intensities were used to fit melting curves and calculate the melting temperature $T_m$ of each protein separately in control (vehicle only) and compound-treated conditions. **(B, C, D)** Melting curves for selected kinases, signaling molecules, and apoptosis regulators showing $T_m$ shift (positive) after treatment with longdaysin (B), roscovitine (C), and SP600125 (D). Data from two independent replicate experiments are shown (please see Figs S10–S12 for further details). **(E, F)** Melting curves for target proteins stabilized by multiple circadian period–altering compounds. Mean values from two independent replicates are represented.

proteins embedded in mammalian circadian feedback loops are vital for period determination and robustness of the clock (Godinho et al, 2007; Shi et al, 2013). Importantly, genetic siRNA knockdown of the target proteins results in long period circadian rhythms, phenocopying the effects of the circadian rhythm–modulating compounds: RABEP1 (26.03 ± 0.92 h), FBX21 (25.96 ± 0.34 h), and CSTF2T (32.85 ± 4.48 h) (Fig S13D). As a further genetic test of the specificity of the circadian rhythm–modulating compounds towards their various targets, we determined their effects on the circadian clock of the prokaryotic cyanobacterium *Synechococcus elongatus*. This organism is the simplest circadian model system and does not possess the target binding proteins that we identified in our integrated proteomics analysis in mammals. Strikingly, we did not find any effect of the compounds in cyanobacteria, indicating their specificity towards the eukaryotic clockwork and to the targets that we identified in human cells (Fig S14). Thus, unbiased proteome-wide profiling of thermal stability delineates a complementary set of molecular targets for circadian rhythm

modulators, alongside novel kinase networks, which were not known previously.

Finally, to obtain a comprehensive depiction of the mechanisms of action and cellular effects of the circadian rhythm–modulating compounds, we investigated the possible interactions among the altered proteins, phosphoproteins, inhibited kinases, and binding targets identified by our multidimensional proteomics analyses. Considering only high confidence interactions (score > 0.8), we identified several overlapping networks (PPI enrichment *P*-value $9.06 \times 10^{-11}$) involving phosphorylation, cell cycle, signaling, protein localization, and regulation of translation, metabolism, and apoptosis (FDR < 0.01) (Fig S15). In addition to circadian timekeeping, the effects of the circadian rhythm–modulating compounds on the cell cycle, cell signaling, and apoptosis fits with the growing links between clock dysregulation and tumorigenesis (Lee et al, 2011; Oshima et al, 2019). Altogether, our results demonstrate that the effects of circadian rhythm–modulating compounds are not restricted to a handful of known targets, but rather that they function

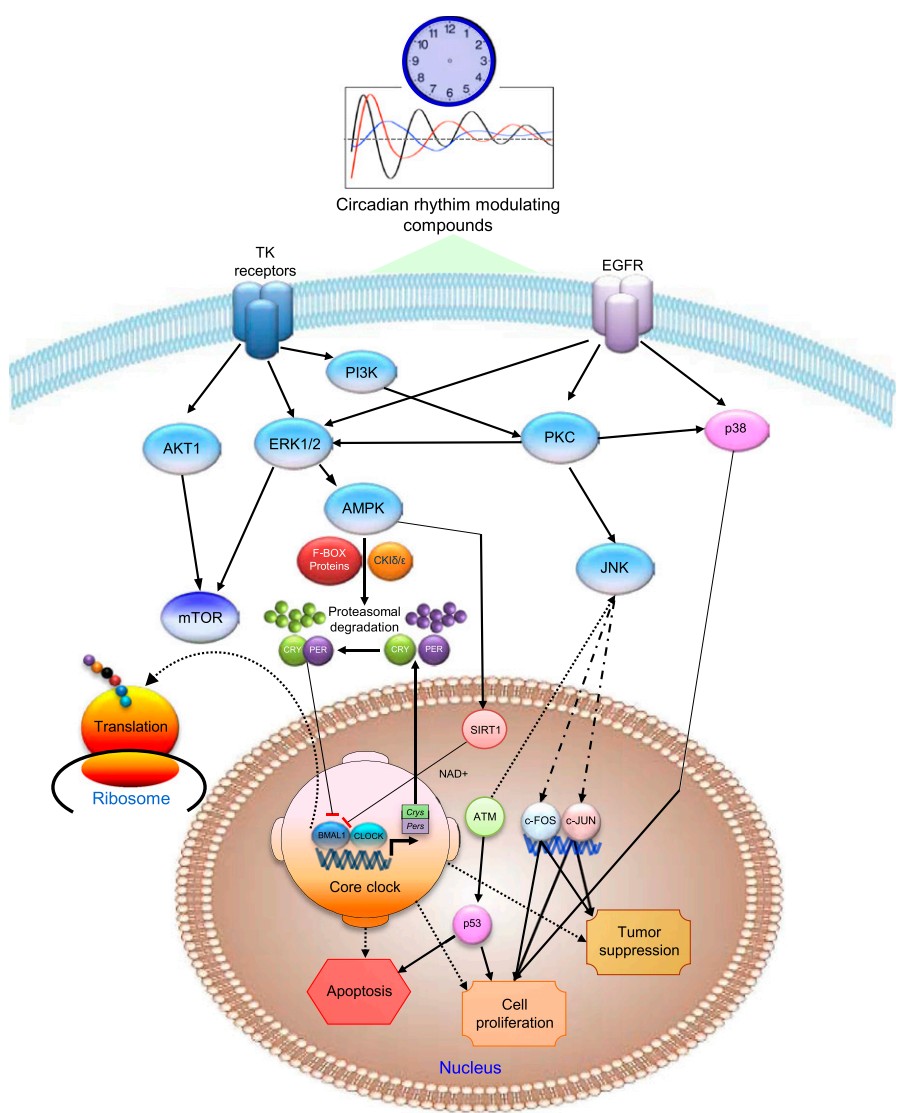

**Figure 7. Schematic depicting the mechanisms of action for circadian rhythm–altering compounds in mammalian systems.**
The major molecular targets and cellular responses induced by these compounds are shown. Circadian period–altering compounds regulate clock machinery through multiple signaling pathways, including MAPK, AMPK, and mTOR, and ubiquitination mediated by F-box proteins. Modulation of circadian timekeeping by these compounds causes alterations in several proteins involved in cell proliferation, apoptosis, and tumor suppression (the schematic diagram is structured using the information obtained from IPA).

by acting on multiple cellular targets embedded within diverse, but well-defined, physiological pathways (Fig 7).

# Discussion

In recent years, systems level approaches have contributed immensely to elucidate complex mechanisms of the circadian clocks (Millius & Ueda, 2017; Rey et al, 2018). Here, we applied PPP to decipher the protein targets of entities that result in the same phenotype. In this case, we used this approach to identify mechanistic targets of circadian–modulating compounds that all lengthen the clock period. We identified many new convergent targets for circadian rhythm–modulating compounds besides their canonical effectors, including various kinases. Most kinase inhibitors that target the ATP-binding site inhibit multiple kinases and even different groups of kinases because of the evolutionarily conserved nature of the binding site (Anastassiadis et al, 2011). Several potential small molecule drug candidates and many approved drugs also have multiple physiological targets, which often cumulatively contribute to a complex mechanism of action (Keiser et al, 2009; Anighoro et al, 2014). In light of this, multi-target activity for most promising circadian period–modulating compounds is not adequately defined (He & Chen, 2016). Here, we have demonstrated that circadian rhythm–modulating compounds with known primary targets also act on several other molecules and downstream pathways to regulate circadian timekeeping (Fig 7). The diversity of targets for circadian rhythm–altering compounds revealed here by PPP provides additional explanations for earlier observations indicating that CK1 d/e-targeting compounds induce much higher period-lengthening effects than CK1d gene knockout (Etchegaray et al, 2009; Isojima et al, 2009) and circadian rhythm–modulating compounds such as longdaysin cause period lengthening even in CK1d deficient cells, that is, in the absence of its primary target (Hirota et al, 2010).

Importantly, a number of regulatory components of the clock machinery are thought to also function beyond their role in circadian rhythmicity. As such, we observed that clock modulator compounds regulate diverse physiological pathways alongside biological rhythms, including protein translation, posttranslational modifications, metabolism, cell proliferation, and apoptosis. We found that circadian rhythm–modulating compounds directly regulate a multi-kinase network mainly revolving around CK1, ERKs, CDKs, TNIK, and STK26. Protein kinases regulate functions of numerous eukaryotic proteins (Manning et al, 2002) and are, therefore, fundamental to cellular and organismal function (Chang & Karin, 2001; Saxton & Sabatini, 2017). Moreover, protein kinases are currently one of the most important group of pharmacological targets because irregular phosphorylation is a cause or consequence of a significant number of human diseases (Cohen, 2002; Ferguson & Gray, 2018; Steinberg & Carling, 2019). Consequently, circadian period or phase-altering compounds that act on multi-kinase targets to modulate daily rhythms are promising broadspectrum drug candidates.

The convergent targets for circadian rhythm–modulating compounds identified here augment our understanding of their mode of action, whereas their effects on different physiological pathways will aid the elucidation of off-targets and possible side effects of these compounds and their future derivatives. A recent study demonstrated the multi-target effects of NCC007, a newly developed compound with structural similarity to longdaysin, which is also a potent period-lengthening compound (Lee et al, 2019). The kinase targets for NCC007 and longdaysin are similar (CK1d, CK1α, CDK 2/7, and MAPK1), suggesting that once all possible targets for these compounds are adequately characterized and more efficient new compounds could be synthesized through structural modifications. However, it is imperative to understand the cellular effects induced by these drug-like compounds to define their possible side effects, because drug discovery to approval and clinical translation to the market is an extremely stringent procedure with a high rate of failure due to off-target effects. In this regard, it will also be important to investigate circadian clock modulators from existing approved drugs based on the targets and pathways identified in our study. For instance, dehydroepiandrosterone (DHEA) was identified earlier as a circadian period–shortening compound through screening approved drug compounds (Tamai et al, 2018), and similar approaches could identify drugs for repurposing as circadian-active agents.

In summary, we provide here a comprehensive atlas for the mechanisms of action and cellular responses for circadian period–lengthening compounds by using PPP. We also demonstrate dose-dependent phase-shifting effects of a few kinase inhibitors. Surprisingly, we found that the mechanism of circadian rhythm modulation is not restricted to previously defined primary targets, but rather that a multi-kinase network is involved. The compounds studied interact with the clock machinery through diverse signaling pathways including MAPK, AMPK, and mTOR. Importantly, serine and threonine kinases (MAPKs and CKs) are in general highly conserved in eukaryotes. We also found ubiquitination mediated by an F-box protein (FBX21) as a common target for these compounds. It remains to be seen if these compounds induce dose-dependent period and phase shifts in diverse types of eukaryotic organisms by regulating similar mechanisms, which could be an interesting continuation of the present study. Collectively, we anticipate that our findings will provide new opportunities for developing novel therapeutics targeting circadian rhythms. More generally, the PPP approach established here provides a paradigm for numerous potential other phenotype-centric molecular analyses. For example, PPP could easily be applied to determine de facto cellular effectors for cell cycle modulators (for cancer drugs) or CRISPR activators and inhibitors (for modulating genome editing) in any model system ranging from bacteria to humans to dissect their mechanisms of action in unprecedented detail.

# Materials and Methods

### Cell culture

*Per2-dLuc* U2OS osteosarcoma and *Bmal1-dLuc* U2OS osteosarcoma reporter cell lines were gifts from Dr Andrew Liu, University of Memphis, USA (Liu et al, 2008). Wild-type U2OS (ATCC-HTB-96), *Per2-dLuc* U2OS, and *Bmal1-dLuc* U2OS cells were grown in DMEM

containing 4.5 g/l glucose supplemented with 10% (vol/vol) new-born calf serum, 1% (vol/vol) GlutaMAX supplement, 1% (vol/vol) penicillin–streptomycin, and 0.2% (vol/vol) MycoZap. The cells were cultured at 37°C, 5% $CO_2$ in a standard humidified incubator.

$S.$ $elongatus$ (strain AMC149) was a gift from Dr Carl Johnson, Vanderbilt University, USA (Kondo et al, 1993). Cyanobacteria cells were grown in BG-11 medium with aeration in constant light (LL) of 100 $\mu$E s$^{-1}$ m$^{-2}$ at 30°C as described earlier (Edgar et al, 2012). The $D_{730nm}$ was maintained between 0.27 and 0.45 by dilution with fresh medium. The cultures were given a 12-h dark pulse to synchronize the clocks, and then returned to constant light.

### Cell-based circadian luminescence assays

To perform bioluminescence recordings, $Per2-dLuc$ U2OS and/or $Bmal1-dLuc$ U2OS cells were grown to confluence and were synchronised by changing the medium to "air medium" (Hastings et al, 2005)—DMEM supplemented with 5 g/l glucose, 20 mM Hepes, 1% penicillin–streptomycin, 0.035% NaHCO3, 10% newborn calf serum, 1% GlutaMAX, 0.5% B-27 Supplement, 1 mM luciferin, 1% non-essential amino acids, and 0.2% MycoZap Plus-PR. Circadian period–altering compounds (dissolved in DMSO, final 0.5% DMSO) were then added in the air medium to reach the different desired concentrations. Bioluminescence assays were performed following the same protocol as described previously (Rey et al, 2016). In brief, the assays were performed at 37°C using 384-well plates in custom-made bioluminescence recording systems (Cairn Research Ltd) composed of a charge-coupled device camera (Andor iKon-M 934) mounted on the top of an Eppendorf Galaxy 170R $CO_2$ incubator. Images were generated from integrated photon counts over 25 min every 30 min using Metamorph software (Molecular Devices). For bioluminescence recordings in $S.$ $elongatus$ (AMC149), $psbAI$p:$luxAB$ was used as a reporter, using earlier described protocols (Edgar et al, 2012). Exported images were composed into image stacks and were quantified in a time series using a custom script in NIH ImageJ software, applying the "Multi Measure" plugin. A modified version of the R script "CellulaRhythm" (Hirota et al, 2008) was used to analyse bioluminescence data traces, and period analysis was performed by sine-wave fitting.

### MS sample preparation

Confluent U2OS cell cultures grown in 150-mm dishes (2 × 10$^7$) were treated with circadian period–altering compounds (longdaysin [cat. no. SML0127; Sigma-Aldrich], purvalanol A [cat. no. P4484; Sigma-Aldrich], roscovitine [cat. no. 557360; Calbiochem] and SP600125 [cat. no. S5567; Sigma-Aldrich]) to a final concentration of 10 $\mu$M dissolved in DMSO (100 $\mu$l of DMSO; <0.5% of the total culture volume) or an equivalent amount of DMSO (vehicle only). After 48-h incubation with the compounds or DMSO, the cells were harvested by trypsinization, washed twice in PBS, and were lysed in 500 $\mu$l lysis buffer containing 50 mM Hepes (pH 8.5), 8M urea, 1% NP-40, protease and phosphatase inhibitors, and benzonase nuclease. Then mild sonication was applied for 15 min (30 s on, 30 s off; medium power) using a Bioruptor Standard (Diagenode) instrument and lysates were centrifuged at 16,000$g$ for 20 min at 4°C. Supernatants were carefully separated and transferred into new microcentrifuge

tubes. Protein precipitation was performed with 1:6 volume of prechilled (–20°C) acetone overnight at 4°C. After overnight incubation, the lysates were centrifuged at 14,000$g$ for 15 min at 4°C. Supernatants were discarded without disturbing pellets, and the pellets were air-dried for 2–3 min to remove residual acetone. Then the pellets were dissolved in 500 $\mu$l 100 mM triethylammonium bicarbonate (TEAB) buffer.

### Multiplexed quantitative proteomics and phosphoproteomics

Protein concentration in each sample was determined by Pierce BCA Protein Assay Kit (cat. no. 23225; Thermo Fisher Scientific) following the manufacturer's instructions. 300 $\mu$g protein per condition was transferred into new microcentrifuge tubes and 15 $\mu$l of the 200 mM tris(2-carboxyethyl)phosphine (TCEP) was added to reduce the cysteine residues, and the samples were incubated at 55°C for 1 h. Subsequently, the reduced proteins were alkylated with 375 mM iodoacetamide (freshly prepared in 100 mM TEAB) for 30 min under dark condition at room temperature. Then, trypsin (Trypsin Gold, Mass Spectrometry Grade; cat. no. V5280; Promega) was added at a 1:40 (trypsin: protein) ratio and the samples were incubated at 37°C for 12 h for proteolytic digestion. After in-solution digestion, control (DMSO) and circadian period–altering compounds treated samples were labelled with TMT isobaric label reagents (TMT, 10-plex) following the manufacturer's instructions (cat. no. 90113; Thermo Fisher Scientific) (Fig 2A). Peptide samples were labelled with the corresponding TMT 10-plex reagent for 1 h 30 min at room temperature. The reactions were quenched using 5 $\mu$l of 5% hydroxylamine for 30 min and 10% of the TMT-labelled peptides were used for global proteomics analysis, whereas the remaining 90% was subjected to phosphopeptide enrichment. Phosphopeptides enrichment was performed using Pierce TiO$_2$ Phosphopeptide Enrichment and Clean-up Kit (cat. no. A32993; Thermo Fisher Scientific) as per the manufacturer's instructions. TMT-labelled peptides were desalted using C18 Spin Tips, dried by vacuum centrifugation, and stored at –80°C until mass spectrometric analysis.

### Kinase profiling using desthiobiotin nucleotide probes

U2OS cells (2 × 10$^7$) were lysed using Thermo Fisher Scientific Pierce IP Lysis Buffer and desalted using Thermo Fisher Scientific 7K Zeba Spin Desalting Columns following the manufacturer's protocol (Pierce Kinase Enrichment Kit with ATP Probe, cat. no. 88310). Cell lysates (1 mg in 500 $\mu$l Reaction Buffer) were treated with 10 $\mu$l of 1M MnCl$_2$ for 1 min at room temperature. Subsequently, the cell lysates (1 mg) were treated with 25 $\mu$M of longdaysin or purvalanol A or roscovitine or SP600125, whereas the control samples were treated with DMSO (vehicle) only (Fig 3A). Compound-treated and control cell lysates were labelled with 5 $\mu$M of desthiobiotin-ATP (Thermo Fisher Scientific) for 10 min at room temperature. Desthiobiotin-ATP labelled proteins were reduced and alkylated before buffer exchange into digestion buffer (20 mM Tris, pH 8.0, and 2M urea). Then the samples were tryptically digested for 4 h; active-site peptides were captured with streptavidin agarose resin and eluted using 50% acetonitrile/ 0.1% TFA. TMT labelling (10-plex) of the kinase-enriched samples was carried out following the same protocol as described above.

## TPP using intact cells

Sample preparation (intact live U2OS cells) for TPP was carried out following the same protocol as described by Franken et al (2015) with slight modifications. In brief, confluent U2OS cell cultures were treated with circadian period–altering compounds (10 $\mu$M of longdaysin, roscovitine, or SP600125) dissolved in DMSO or an equivalent amount of DMSO (vehicle only). Incubation of cells with the compounds and vehicle was conducted in parallel at 37°C. TPP experiments were performed, including two biological replicates for each condition. The cells were washed twice with ice-cold PBS and were harvested by trypsinization after incubation with circadian period–altering drugs. The cells were resuspended in ice-cold PBS containing 0.4% NP-40 and protease inhibitors, and the resulting cell suspension was divided into 10 0.2-ml PCR tubes (50 $\mu$l each). Detergent Nonidet P-40 (NP-40) was included in buffer to detect transmembrane protein–small molecule interactions (Reinhard et al, 2015). Compound- and vehicle-treated cells were then heated in parallel in a PCR machine for 3 min to the respective temperature (10 temperatures: 37°C, 41°C, 44°C, 47°C, 50°C, 53°C, 56°C, 59°C, 63°C, and 67°C) (Fig 6A), followed by a 3-min incubation at room temperature. Subsequently, the cells were snap-frozen in liquid nitrogen for 1 min and were thawed briefly in a water bath at 25°C and were transferred on ice and resuspended by using a pipette. This freeze–thaw cycle was repeated two times. The entire content was centrifuged at 100,000$g$ for 20 min at 4°C. After centrifugation, 30 $\mu$l of the supernatant was transferred into a new tube. Protein concentration in each sample was determined by Pierce BCA Protein Assay Kit following the manufacturer's instructions. 50 $\mu$g of the lowest temperature sample and an equivalent volume of the other temperature-point samples were analysed using SDS–PAGE for a quality control check. TMT labelling (10-plex) was carried out following the same protocol as described above. TMT-labelled peptides were fractionated using Pierce High pH Reversed-Phase Peptide Fractionation Kit (cat. no. 84868; Thermo Fisher Scientific) following the manufacturer's instructions. All fractions (eight fractions in each set) were desalted using Pierce C18 Spin Tips, dried by vacuum centrifugation and stored at –80°C until mass spectrometric analysis.

## LC-MS/MS analysis

In the integrated quantitative proteomics and phosphoproteomics workflow, the pooled TMT-labelled samples were fractionated using an Ultimate 3000 nanoHPLC system in line with an Orbitrap Fusion Tribrid mass spectrometer (Thermo Fisher Scientific). In brief, peptides in 1% (vol/vol) formic acid (FA) were injected onto an Acclaim PepMap C18 nano-trap column (Thermo Fisher Scientific). After washing with 0.5% (vol/vol) acetonitrile, 0.1% (vol/vol) FA, peptides were resolved on a 250 mm × 75 $\mu$m Acclaim PepMap C18 reverse-phase analytical column (Thermo Fisher Scientific) over a 150-min organic gradient, using six gradient segments (5–9% solvent B over 2 min, 9–25% B over 94 min, 25–60% B over 23 min, 60–90% B over 5 min, held at 90% B for 5 min, and then reduced to 1% B over 2 min) with a flow rate of 300 nl/min. Solvent A was 0.1% FA and solvent B was aqueous 80% acetonitrile in 0.1% FA. Peptides were ionized by nano-electrospray ionization at 2.0 kV using a

stainless-steel emitter with an internal diameter of 30 $\mu$m (Thermo Fisher Scientific) and a capillary temperature of 275°C. All spectra were acquired using an Orbitrap Fusion Tribrid mass spectrometer controlled by Xcalibur software *version* 2 (Thermo Fisher Scientific) and operated in data-dependent acquisition mode using a synchronous precursor selection-MS3 workflow. FTMS1 spectra were collected at a resolution of 120,000, with an AGC target of 200,000 and a max injection time of 50 ms. Precursors were filtered with an intensity threshold of 5,000, according to charge state (to include charge states 2–7) and with monoisotopic precursor selection. Previously interrogated precursors were excluded using a dynamic window (60 s ± 10 ppm). The MS2 precursors were isolated with a quadrupole mass filter set to a width of 1.2 m/z. ITMS2 spectra were collected with an AGC target of 10,000, max injection time of 70 ms and collision-induced dissociation energy of 35%. For FTMS3 analysis, the Orbitrap was operated at 50,000 resolution with an AGC target of 50,000 and a maximum injection time of 105 ms. Precursors were fragmented by high-energy collision dissociation (HCD) at a normalized collision energy of 60% to ensure maximal TMT reporter ion yield. SPS was enabled to include up to five MS2 fragment ions in the FTMS3 scan.

In TPP and kinase enrichment experiments, peptide mixtures from TMT 10-plex labelled samples were chromatographically resolved on an EASY-spray PepMap RSLC C18 column (2 $\mu$m, 100 Å, 75 $\mu$m × 50 cm ID) using an Ultimate 3000 RSLCnano system (Thermo Fisher Scientific) over a 180-min gradient at 40°C. Six gradient segments (2–8% solvent B over 3 min 30 s, 8–25% B over 98 min 30 s, 25–40% B over 45 min, 40–95% B over 5 min, held at 95% B for 5 min, and then reduced to 2% B over 1 min and for the rest of the acquisition) were applied with a flow rate of 250 nl/min. Two gradient segments (2–35% solvent B over 153 min, then increased to 95% in 1 min and held at 95% B for 11 min and then reduced to 2% B over 1 min and for the rest of the acquisition) were applied in TPP experiments. Solvent A was 0.1% FA and 5% DMSO in HPLC-grade water, and solvent B was 0.1% FA and 5% DMSO in 80% acetonitrile. All spectra were acquired using an Orbitrap Fusion Lumos Tribrid mass spectrometer (Thermo Fisher Scientific). Xcalibur (version 2 and 4) software (Thermo Fisher Scientific) was used to control data acquisition. The instrument was operated in data-dependent acquisition mode, and the most abundant peptides selected for MS/MS were fragmented by HCD fragmentation techniques. MS spectra were acquired at a resolution of 120,000, and an ion target of 4E[5]. HCD scans were performed with 38% normalized collision energy at 50,000 resolution (at 100 m/z), and the ion target settings were set to 1E[5]. Precursors were filtered with an intensity threshold of 5,000, according to charge state (to include charge states 2–6) and with monoisotopic precursor selection.

## In vitro kinase assays

The Reaction Biology (http://www.reactionbiology.com) "HotSpot" assay platform was used to validate inhibition of different kinases by circadian period–altering compounds as identified in the proteomic approach using desthiobiotin nucleotide probes. In vitro profiling of selected kinase panels was performed following the same protocol as described by Anastassiadis et al (2011). In brief, specific kinase/substrate pairs were mixed in a buffer containing

20 mM Hepes (pH 7.5), 10 mM $MgCl_2$, 1 mM EGTA, 0.02% Brij35, 0.02 mg/ml BSA, 0.1 mM $Na_3VO_4$, 2 mM DTT, and 1% DMSO. Circadian rhythm–modulating compounds (longdaysin, purvalanol A, roscovitine, and SP600125) were then added to each reaction mixture. After a 20-min incubation, ATP (cat. no. A7699; Sigma-Aldrich) and [γ-$^{33}$P] ATP (cat. no. NEG602H001MC; PerkinElmer) were added at a final total concentration of 10 mM. Reaction mixers were incubated at 25°C for 120 min and spotted onto P81 ion exchange cellulose chromatography paper (cat. no. 05-717; Whatman). Unbound phosphate was removed by washing the filters in 0.75% phosphoric acid for multiple times. After subtraction of background derived from control reactions containing inactive enzyme, kinase activity data were expressed as the percent remaining kinase activity in treated samples compared with vehicle (DMSO) reactions. 10-dose inhibition curves (threefold serial dilution starting at 100 $\mu$M) were used to derive the $IC_{50}$ values and curve fits were obtained using Prism (GraphPad Prism, *version* 8.1.0).

## Database search for peptide and protein identification

Quantitative proteomics and phosphoproteomics, and TPP raw data files were analysed using the MaxQuant computational platform (*version* 1.5.2.8) with the Andromeda search engine (Cox & Mann, 2008). MS2/MS3 spectra were searched against UniProt database specifying *Homo sapiens* (Human) taxonomy (proteome ID: UP000005640; organism ID: 9606; protein count: 73,920). All searches were performed using "Reporter ion MS2/MS3" with "10-plex TMT" as isobaric labels with a static modification for cysteine alkylation (carbamidomethylation) and oxidation of methionine (M) and protein N-terminal acetylation as the variable modifications. Phospho (STY) was included as an additional variable modification for the phosphopeptide enriched samples. Trypsin digestion with maximum one missed cleavage, minimum peptide length as seven amino acids, precursor ion mass tolerance of 5 ppm, and fragment mass tolerance of 0.02 kD was specified in all analyses. The FDR was kept at 0.01 for peptide spectrum match, protein, and site decoy fraction. TMT signals were corrected for isotope impurities based on the manufacturer's instructions.

The raw data files for kinase enrichment analysis were processed and quantified using Proteome Discoverer software *version* 1.4 (Thermo Fisher Scientific) and were searched against the UniProt Human database using the SEQUEST algorithm. Peptide precursor mass tolerance was set at 10 ppm, and MS/MS tolerance was set at 0.06 kD. Search criteria included oxidation of methionine (M), TMT to lysine, and desthiobiotin lysine (K) as variable modifications, whereas carbamidomethylation of cysteine and TMT to peptide N termini were included as fixed modifications. Searches were performed with full tryptic digestion and a maximum of one missed cleavage was allowed. All peptide data were filtered at high (1%) and medium (5%) FDR.

## Statistical analysis of quantitative proteomics and kinase assay data

Unless otherwise mentioned, all experiments were conducted with biological replicates and at least two independent experiments were performed. Processing and statistical analysis of quantitative proteomics and phosphoproteomics datasets were performed using the Perseus workstation (*version* 1.5.5.3) (Tyanova et al, 2016). During data processing, reverse and contaminant database hits and candidates identified only by site were filtered out. In differential quantitative proteomics analyses (control versus compound-treated groups), categorical annotation was applied to group reporter ion intensities, values were log2 transformed, and were normalized by "subtract mean (column-wise)" in each TMT reporter ion channel. Protein groups were filtered for valid values (at least 80% in each group), and *P*-values obtained from a paired *t* test were used to determine the significance of differences in protein or phosphosite abundances between control and compound-treated groups. *P*-value (adjusted) < 0.05 were considered as statistically significant.

GraphPad Prism 8 and XLSTAT were used for performing statistical tests for kinase assay data. Succeeding one-way ANOVA, Dunnett's test was performed to compare every (compound-treated groups) mean to a control (DMSO) mean. *P*-value < 0.05 was considered statistically significant. Venny *version* 2.1 was applied to generate the Venn diagrams for detecting overlaps among different conditions.

## TPP data analysis

In TPP quantitative proteomics experiments, fold-changes were calculated by using the lowest temperature condition (37°C) as the reference. These fold-change values indicate the relative amount of nondenatured protein at the corresponding temperature. Curve fitting (relative fold-changes as a function of temperature), normalization, and assessment of slope and melting point differences were performed using the TR workflow of the TPP R package (Franken et al, 2015). Subsequent to a global normalization procedure (Savitski et al, 2014), melting curves were fitted for all proteins identified in the vehicle- and compound-treated experiments. Melting point differences and slope differences were calculated for proteins for which the melting curves passed the following two requirements: (i) both fitted curves for the vehicle- and compound-treated condition had an $R^2$ of greater than 0.8, and (ii) the vehicle curve had a *plateau* of less than 0.3. Subsequently, four filters were used to determine the quality of the result for each protein. (i) Minimum slope in each of the control versus treatment experiments <−0.06. (ii) Both the melting point differences in the control versus treatment experiments greater than the melting point difference between the two untreated controls. (iii) One of the *P* values (FDR corrected) for the two replicate experiments <0.1. (iv) Melting point shifts in both the control versus treatment experiments have a positive sign (i.e., protein was stabilized in both cases). Candidates satisfying all these selection criteria were considered as the potential cellular targets for circadian period–altering compounds.

## Interaction network and bioinformatics analysis

Functional annotation and clustering of the differentially abundant proteins and phosphosites and altered kinases identified in circadian rhythm–modulating compound–treated cells were performed using Protein ANalysis THrough Evolutionary Relationships (PANTHER) classification system, version 12.0 (Mi et al, 2013), IPA (QIAGEN), and Reactome pathway Knowledgebase, version 62 (Fabregat et al,

2016). Statistical overrepresentation test was performed to select the Gene Ontology terms overexpressed (FDR < 0.05) for the differentially abundant candidates. In addition, Proteomaps were used to assign proteins to functional categories (KEGG Pathways gene classification) (Liebermeister et al, 2014). Protein association networks were generated using the STRING database, version 11.0 (Szklarczyk et al, 2019) involving only high-confidence (score > 0.8) interactions. siRNA knockdown data for the differentially abundant proteins, altered kinases, and potential cellular targets for circadian period–altering compounds were obtained from BioGPS circadian layout database (Zhang et al, 2009). Kinome tree visualization and annotation were performed using KinMap web-based tool (Eid et al, 2017).

### Data availability

The MS proteomics, phosphoproteomics, kinome profile, and TPP data described in this article are deposited to the ProteomeXchange Consortium via the PRIDE (Perez-Riverol et al, 2019) partner repository with the dataset identifier PXD014186.

# Supplementary Information

# Acknowledgements

AB Reddy acknowledges funding from a Wellcome Trust Senior Fellowship in Clinical Science (100333/Z/12/Z), the European Research Council (Starting Grant No. 281348, MetaCLOCK), the European Molecular Biology Organization Young Investigators Programme, the Lister Institute of Preventive Medicine, and the University of Pennsylvania Perelman School of Medicine. This work was supported in part by The Francis Crick Institute, which receives its core funding from Cancer Research UK (FC001534), the UK Medical Research Council (FC001534), and the Wellcome Trust (FC001534). S Ray acknowledges his Thermo Scientific Tandem Mass Tag Research Award (2018). Support provided by the Reaction Biology Corporation (Malvern, PA) in radiometric "HotSpot" assays is gratefully acknowledged.

## Author Contributions

S Ray: conceptualization, data curation, formal analysis, investigation, visualization, methodology, and writing—original draft, review, and editing.
R Lach: formal analysis, investigation, and visualization.
KJ Heesom: formal analysis, investigation, and methodology.
UK Valekunja: formal analysis, investigation, visualization, and methodology.
V Encheva: investigation and methodology.
AP Snijders: methodology.
AB Reddy: conceptualization, resources, formal analysis, supervision, funding acquisition, visualization, methodology, project administration, and writing—original draft, review, and editing.

## Conflict of Interest Statement

The authors declare that they have no conflict of interests.

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
