## [Reviewer comments · Life Science Alliance]

Proteomic Profiling Identifies a Landscape of Targets for Circadian Clock Modulating Compounds

Sandipan Ray, Radoslaw Lach, Kate J. Heesom, Utham K. Valekunja, Vesela Encheva, Ambrosius P. Snijders, and Akhilesh B. Reddy

DOI: 10.26508/lsa.201900603

Corresponding author(s): Dr. Akhilesh Reddy (University of Pennsylvania)

Review timeline:

Submission Date:	2019-11-08
Editorial Decision:	2019-11-08
Revision Received:	2019-11-18
Editorial Decision:	2019-11-18
Revision Received:	2019-11-19
Accepted:	2019-11-19

Scientific Editor: Andrea Leibfried

Transaction Report:

Please note that the manuscript was previously reviewed at another journal and the reports were taken into account in the decision-making process at Life Science Alliance.

No Peer Review Process File is available with this article, as the authors have chosen not to make the review process public in this case.

1st Editorial Decision

08 November 2019

Re: Life Science Alliance manuscript #LSA-2019-00603-T

Prof Akhliesh Reddy
The Francis Crick Institute
London

Dear Dr. Reddy,

Thank you for transferring your manuscript entitled "Phenotypic Proteomic Profiling Identifies a Landscape of Targets for Circadian Clock Modulating Compounds" to Life Science Alliance. The manuscript was assessed by expert reviewers at another journal before, and the editors transferred those reports to us with your permission.

The reviewers who evaluated your work at the other journal appreciated the quality of the data, but would have expected a broader conceptual advance. Lack thereof does not preclude publication in Life Science Alliance in this case, and we would thus like to invite you to submit a revised version for publication here.

Please provide a point-by-point response to the concerns raised and introduce text changes in your manuscript accordingly. The following editorial points should get addressed, too:

- Please link your profile in our submission system to your ORCID iD, you should have received an email with instructions on how to do so
- Please provide a short summary blurb about your work in our submission system and fill in the other mandatory fields
- Please upload all figure files (including supplementary figures) as separate files and without legends, the latter should only remain in / get added to the manuscript word docx file
- Please make sure that all figures are easily readable - Parts of the PPP graph in figure 1 are a bit blurry and figure S10 is really dense at the moment and could perhaps be split into two suppl figures (note that all figures, including suppl. Figures will be displayed in-line in the HTML version of your paper)

To upload the revised version of your manuscript, please log in to your account: <https://lsa.msubmit.net/cgi-bin/main.plex>
You will be guided to complete the submission of your revised manuscript and to fill in all necessary information. Please get in touch in case you do not know or remember your login name.

Thank you for this interesting contribution to Life Science Alliance. We are looking forward to receiving your revised manuscript.

Sincerely,

B. MANUSCRIPT ORGANIZATION AND FORMATTING:

RE: Life Science Alliance Manuscript #LSA-2019-00603-TR

Dr. Akhliesh Reddy
University of Pennsylvania
Smilow Center for Translational Research
Philadelphia 19104

Dear Dr. Reddy,

Thank you for submitting your revised manuscript entitled "Proteomic Profiling Identifies a Landscape of Targets for Circadian Clock Modulating Compounds". I appreciate your response to the concerns raised and would be happy to publish your paper in Life Science Alliance.

Please still include a reference in the figure legend to either Fig 6B-D or S10-S12 that the same is shown as in the respective other figure.

A. FINAL FILES:

B. MANUSCRIPT ORGANIZATION AND FORMATTING:

Sincerely,

Andrea Leibfried, PhD
Executive Editor
Life Science Alliance
Meyershofstr. 1
69117 Heidelberg, Germany
t +49 6221 8891 502
e a.leibfried@life-science-alliance.org
www.life-science-alliance.org

3rd Editorial Decision

19 November 2019

RE: Life Science Alliance Manuscript #LSA-2019-00603-TRR

Dr. Akhliesh Reddy
University of Pennsylvania
Smilow Center for Translational Research
Philadelphia 19104

Dear Dr. Reddy,

Thank you for submitting your Research Article entitled "Proteomic Profiling Identifies a Landscape of Targets for Circadian Clock Modulating Compounds". It is a pleasure to let you know that your manuscript is now accepted for publication in Life Science Alliance. Congratulations on this interesting work.

DISTRIBUTION OF MATERIALS:

Again, congratulations on a very nice paper. I hope you found the review process to be constructive and are pleased with how the manuscript was handled editorially. We look forward to future exciting submissions from your lab.

Sincerely,
